# SPIDER as A Rehabilitation Tool for Patients with Neurological Disabilities: The Preliminary Research

**DOI:** 10.3390/jpm10020033

**Published:** 2020-04-30

**Authors:** Sebastian Glowinski, Andrzej Blazejewski

**Affiliations:** Koszalin University of Technology, Faculty of Mechanical Engineering, Department of Mechatronics and Automatic, Sniadeckich 2, 75-453 Koszalin, Poland; andrzej.blazejewski@tu.koszalin.pl

**Keywords:** SPIDER therapy, rehabilitation, computer modelling, vertical strength model

## Abstract

(1) Background and purpose: SPIDER (Strengthening Program for Intensive Developmental Exercises and activities for Reaching health capability) is dedicated for patients suffering from Cerebral Palsy, Sclerosis Multiplex, Spinal Bifida, Spinal Muscular Atrophy and strokes. Authors proposed a computer model for the evaluation patient’s condition and the rehabilitation progress. (2) Methods: The 2-year-old and 76-year-old patients with neurological problems, who underwent individual therapy included balancing and coordination practising with SPIDER device. The model comparing the forces, which act during the therapy process, such as the expander and gravity forces, was worked out using Matlab software. (3) Results: The model allowed controlling the changes into the patients centre of gravity forces continuous adjustment and postural stability during any patient’s movement. After rehabilitation sessions, lasted for 28 days during which patients received the progress information and the therapist got the numeric data, regarding the period of the therapy. (4) Conclusions: The first patient was able to move, dramatically improved the ability to balance and coordination. The second one presented change in gait, improvement in mobility, motor function and decreased fall risk. The proposed computer model gives information about the forces acting to the patient body. The physiotherapist can evaluate the progress of patient verticalization and receive information, in the form of numbers and charts.

## 1. Introduction

Stroke is the main cause of adult long-term disability, frequently leading to significant gait and trunk control impairment [1]. Rehabilitation is an important part of recovery after stroke. There are many approaches to stroke rehabilitation. One of them is physical activity. For example, motor-skill exercises can help improve patient muscle strength and coordination [2]. In this part of rehabilitation physical therapists are involved. They help relearn movements such as walking and keeping balance [3]. Early mobilization and postural changes are improving physical and psychological outcomes [4]. Patients, with disabilities caused by a stroke, can effectively improve their ability to walk and carry out daily activities introducing dedicated exercises and training. Exercises are aimed to increase muscle strength, improve fitness and as the final result to enhance general health [5]. For example, symmetrical body-weight distribution training may improve sit-to-stand performance and consequently decrease the number of falls in case of patients stroked out [6,7].

The SPIDER Strengthening Program for Intensive Developmental Exercises and activities for Reaching health capability) system came into being in 1993 and has helped to improve a mobility and independence of patients with neurological disorders. Most of the patients trained in SPIDER device suffered from Cerebral Palsy, Sclerosis Multiplex, Spinal Bifida, Spinal Muscular Atrophy (SMA) and strokes. Many specialists have appreciated the advantages of this equipment. It has been tested in a Public Open Treatment Hospital in Warsaw and also in the area of Warsaw. SPIDER can be used for the therapy of both children and elder people. A therapist leads a rehabilitation program with full consent of the patients. SPIDER equipment was developed as an answer to the missing link in rehabilitation [8]. It enables physicians and therapists to design and administer a complete range of rehabilitation for almost every type of disorder [9]. Exercises with SPIDER cage can strengthen patients’ muscles, improve their coordination, help in verticalization and in finding body balance by using very simple but very usable body supporting elements, described further. Consequently, the cage improves the movement of patients wheelchaired and even enables them stand independently.

SPIDER improves patients’ mobility and effectively reduces the disorders of the Central Nervous System, which is beyond a traditional therapy [10,11]. SPIDER device interacts through proprioceptive and incomplete input information. This allows to adjust muscle tone during a movement, maintain a certain position of the body, as well as to keep a sense of balance and confidence to the force of gravity. A change in the muscle tension distribution, as the result of the use of the device, enables the change of the arrangement of individual body parts and completely modifies the image of proprioceptive information [12,13].

The ADELI Suit (loading and training device) an alternative to SPIDER system, was invented in 1991 [14]. The theory, behind this kind of systems, is that through an active movement therapy, the brain is stimulated and thus retrained to recognize, and eventually initiate correct movements of muscles action [15]. 

The following article is the first describing the patient’s condition before and after the therapy in using SPIDER device. Describing the patient’s’ condition, many variables and factors are affecting the therapy impact assessments [16,17]. Patients behave differently during the initial examination then after a certain time of the therapy. Therefore, therapy assessments and scales used are unreliable. Often, there is a so-called white uniform/smock effect. The patient’s condition is subjective. Lindvall proposed six-spot step test with persons after a stroke [18]. This method can be a complementary measure of gait and balance in the stroke victim’s rehabilitation. Roy described, that the asymmetrical motor pattern of persons with hemiparesis, influences the performance of activities that require interactions between the two sides of the body asymmetry. He recorded forces under patient feet using two force plates and under thighs with an instrumented chair [19]. 

We propose another, the new method, based on the specific elastic cord or otherwise expander known characteristics. Two research questions were put forward in this study:-what are the expander characteristics used in the SPIDER cage?-what do the value of the forces act to the patient body in the function of the expander angle to the transverse plane?

## 2. Materials and Methods

The idea of SPIDER equipment, as a part of the entire system (metal cage), is based on elastic cords which are the stimulative elements. The cords are fixed to a carrying belt (carrying strap) attached to the patient’s waist. The generated force depends on the type of expander (two types of cords are used) and the height of attachment on SPIDER device cage (the height of fixing expander or cord—the centre of gravity—level angle, shown in Figure 1).

The benefit of SPIDER is a possibility to verticalize a patient. It is important when he can’t stand on his own. The patient is unloaded when the cord attachment point to the cage is higher than the waist belt (above the transverse plane and in the same way above the centre of gravity). The patient is loaded If it is fixed lower (under the transverse plane). This quick way of changing load and unload is a useful possibility in the rehabilitation process. So far, the physiotherapist, who used SPIDER device in the therapy, relied on his own experience. Specialists individually determined the value of the load (unload) force, which depended on the condition of the patient and evaluated the therapy progress subjectively.

Therefore, the computer model feasibly properly simulating SPIDER device functionality was created. The proposed model allows to determining the exact value (magnitude) and direction of forces depending on the anthropometric parameters of the patient and the height of the expander assembling. By using the proposed model, it is possible to calculate the real force effects on a particular part of the human body, during every period of the rehabilitation process. In practice, SPIDER device can be used, when the maximum mass of the patient does not exceed 110 kg. The real utility range of the device is 70–80 kg.

In the first step of the model creation, the elastic cords characteristics were determined. It was done by using a dynamometer. Each cord was measured before the study (straight-line length, when the force was 0 N). Then the expanders were stretched (every 0.05 m) and the force was measured. The length of thick elastic cords (Tk) were from 0.605 m to 0.630 m. The length of thin expanders (Tn) were from 0.620 m to 0.655 m. Next, the elongation was calculated as [%] of an initial length. After that, the cord characteristics were calculated by using Matlab Curve Fitting Toolbox (R-square–0.9982 thick and 0.9926 thin expanders) [20]. R-squared is a statistical measure of how close the data are to the fitted function line. The cord force models are based on the sum of sine (1) and (2). The number of factors in the series is 3.
(1)$$T_{n}\left[N\right]=a_{C1}sin\left(b_{C1}x+c_{C1}\right)+a_{C2}sin\left(b_{C2}x+c_{C2}\right)+a_{C3}sin\left(b_{C3}x+c_{C3}\right)$$
(2)$$T_{k}\left[N\right]=a_{G1}sin\left(b_{G1}x+c_{G1}\right)+a_{G2}sin\left(b_{G2}x+c_{G2}\right)+a_{G3}sin\left(b_{G3}x+c_{G3}\right)$$
where:
$a_{C1},\ldots ,c_{C3}$—a thin elastic cord coefficient (with 95% confidence bounds), $a_{G1},\ldots ,c_{G3}$—a thick elastic cord coefficient (with 95% confidence bounds), x—an elongation from 0 to 100 [%].

The cord force characteristics, presented in Figure 2, describe expander real dynamic properties. It is possible to use two expanders (or more) together (parallel) depending on the physiotherapist’s needs. The flexible connectors help the patient to improve his balance and posture, allow to develop self-acting movements and support a movement with greater precision. The calculated models were applied to the Matlab script, which contained the individual anthropometric data of each patient (mass, height, and length of individual body parts). In the Matlab script, the model reduced the forces to two forces attached to two essential human body points. These are the centre of gravity (COG) as well as a foot ground contact (FGC).

On the basis of the data such as the age of the patient, Body Mass Index, type of disease, time since the stroke and its extent, the physiotherapist determines the rehabilitation program. Usually, this is done in the so-called time units. The patients are verticalized in a device repeatedly for 2–3 min, then they get into a comfortable sitting position. The maximum duration of the therapy unit is up to 45–60 min during one session.

## 3. Preliminary Research—Cases Description

The first patient—patient A was a 76-year-old male. He was 1.75 m tall and weighed 75 kg, it means ‘healthy’ BMI of 26.1. He was admitted with the left hemisphere ischemic stroke, which was confirmed by a deep right-sided paresis and aphasia. During the structural assessment on the day of admission to the rehabilitation, the following were observed: a significant weakening of the abdominal and lateral oblique muscles on the right side. Additionally a deep paresis of the right lower and upper limb as well as weakness and decreased postural muscle activity in the torso were also observed. Symptoms of the disease were particularly noticeable as the limited possibilities of taking the vertical position and maintain the posture. It means that before starting the rehabilitation, the patient did not take an upstanding position himself. Due to paresis of the right limb, the patient was not able to move this side independently. 

The first step during the rehabilitation was an attempt to support the patient in taking the most optimal and active upright position using SPIDER capabilities. This was performed to stimulate the equivalent reactions, mutual control and cooperation of antigravity muscles.

The second step was started from the slow and gradual weighting of a right limb with the use of SPIDER device. The appropriate strength of the abductor and external rotators of the hip joint, knee extensor muscles and the three-headed calf were needed to maintain the proper position of the joints during the successive load. It was also expected to resolve the eccentric work of the three-headed calf.

Next, to improve the work of transporting the numb limb, the device was used to stimulated: right hip flexors, the concentric work of the three-headed calf and the trunk to have shortening ability. SPIDER device was used to actively stabilize the right knee joint as much as possible, in combination with the pelvic stabilization using elastic cords properties. It allowed working on the load phase of the affected right limb.

The second patient—patient B was a 2-year-old boy (weight = 12 kg, 0.88 = m, BMI = 15.6). The patient suffered from cerebral palsy in the form of hemiparesis. Although the patient was able to keep standing upright without problems, when he lowered the support plane, he was losing control of the stable posture and overcorrecting One of the most visible problems was abnormal propulsion (transporting) of the foot during the weighing phase of the right lower limb. It was caused by the increased tension of the soleus muscle of the calf and lack of full control over its eccentric activity. It led to its structural shortening and limited possibility of active relaxation. Weak postural muscle activity within the torso and the iliac rim, especially at positions with a reduced support plane was observed. In the rehabilitation process, the proper stabilization was ensured allowing the boy’s nervous system suitable conditions for applicable response and adaptation to working in a situation of limited support plane. These exercises were designed to improve the equivalent reactions while moving the centre of gravity of the body. To reduce these deficits, techniques of muscle mobilization were used. Moreover, attention was focused on the muscle’s eccentric work and the antagonistic muscle group of the ankle dorsiflexes, striving to stimulate the active relaxation. To this end, using the tension of tendons, the resultant force was directed to the unaffected side. Increasing or decreasing the tension of tendons by 10–20%, followed by manual physiotherapist’s treatments, helped to stimulate the patient’s defence reactions to maintain the balance.

## 4. Results

Patient A, in addition to traditional physiotherapeutic treatments, underwent one-hour therapy in SPIDER device twice a day for four weeks. Patient A made significant progress in the field of walking re-education and changing the position from lying down to sitting and from standing to sitting down. On the day of the admission to rehabilitation, he couldn’t take an independent standing position, but before leaving he began to move independently on the tripod, with belaying. For patient A, the approximate progression is 70–80%. However, the continuation of the therapy to maintain the existing functional possibilities and their further improvement is necessary. 

Patient B, similarly as the previous one, had a traditional physiotherapeutic treatment, underwent one-hour therapy in SPIDER device twice a day for four weeks. The approximate progress is about 60% at the end of the therapy. After the therapy, the improvement is still going on, but the lack of continuation of the therapy in SPIDER to consolidate the effects or abandoning the exercises may lead to regression. In the case of long breaks in the therapy in SPIDER device, even 20% regression may occur comparing to the effects obtained in the therapy session. 

Using the computer model, the simulation was done based on two cases considered. The forces coming from elastic cords, the results related to body weight located in the centre of gravity point and reaction induced on the feet and ground contacts are shown in the following figures. In these figures, the human is facing straight towards the surface YZ. This direction is related to the Frontal Plane shown in Figure 1.

The model showed that patient A to take the vertical position and to stimulate torso muscles, a high amount of the patient’s mass had to be suspended (Figure 3a). In the case of an adult, it is possible to reduce the weight approximately by 15% generating symmetric resultant in direction Z and symmetric feet load, similarly to the first younger patient case, but with much higher force values (Figure 3b).

In the next step, while sustaining the rest of the body (Figure 4a), the resultant acting on the right foot stimulates the muscles. Increasing the component X in this point forces patient’s A calf muscle action (Figure 4b). In the case of patient B, to improve the equivalent reactions while moving the centre of gravity, the elastic cord was set in such a way to obtain the resultant in direction Z (Figure 4a). In the first step, this particular elastic cord configuration results in resting patient B in the amount of 16%. Both sides were subjected to the same forces. In the second step, the additional patient’s problem was considered, i.e., the resultant was localized as in Figure 4b. While left foot was forced in the heel area, the right one was stimulated to get the ankle active relaxation. Figure 4a shows that the forces acting in the centre of gravity cause evenly load (equally on both sides, both legs) or particular human side engagement. The resultant (black arrows) in direction Z unweight (relieve) the human body in general, whereas resultant in directions X and Y together loads the particular side respectively. It results in the fact that the symmetrical force configuration, shown in Figure 4b, acts on feet, unweights (relieve) and keeps them together. The Right_y_ and Left_y_ forces (see: Figure 4b) stabilize the body position.

To allow the therapist imposing the patient’s proper force configuration, the right tension which means appropriate length, which leads to suitable fastening points location of the cords should be chosen. It means that unweighting or loading a leg(s) depends on force value. In the same way, the therapist can set up the proper part of the patient’s feet load, shown in Figure 5b. Measuring the forces acting in point FGC (Left or Right - black arrows), it is possible to estimate the forces necessary to impose the load on a heel or toe mound. The forces in direction Z (Left_z_ or Right_z_) unweight (positive force value) or load (negative force value) a heel whereas the forces in direction Y move (positive) or remove (negativity) the load from toe mound.

## 5. Discussion

The present study was based on two medical cases. The results obtained in the form of charts and above analyses of the rehabilitation progress by using SPIDER device are promising and encouraging to perform further research. In order to answer the questions that were asked in Introduction, the following effort was made.

The assumptive methodology pointed out that firstly, we defined the cord force characteristics, which describe expander real dynamic properties. It turned out that the characteristics are not linear with three ranges. The most desirable range of cords, elongation is between 20 to 80% (Figure 2). For every type of cords, one would like to use this range can be calculated. Next, these characteristics were implemented in the model considered. Next, based on the model, we proved that it is possible to receive the information about the component forces acting on the patients in the SPIDER. Even the smallest changes in these components are followed by changes in muscle tension throughout the whole body. The model allowed distinguishing forces with components, which were essential in case of the particular disease. Different forces were considered in case of the one patient (Figure 3 and Figure 4) and the second one (Figure 5). The force and their components values, calculated by the model, could have been chosen depending on the particular patient medical condition. The force acting to the patient body can be simply changed (by shortening, lengthening or replacing the expanders). The postural adjustment can sometimes cause tension changes that are invisible to the naked eye. The model was applied at the beginning of the rehabilitation to evaluate the patient’s condition and its progress. The particular patient medical condition, at the start of rehabilitation, is are described by the forces values and configuration are shown thanks to model. It allowed control of changes in the centre of gravity, enforcing continuous postural adjustment during any movement. Moreover, changes in cords tension triggering appropriate muscles’ reactions can control even the smallest changes in the patient’s position. At the end of rehabilitation, the values and configuration of the forces shown in the model gave a measurable tool to make a rehabilitation progress assessment. In from of numbers one has non-judgmental tool for evaluation. The answer to the second question is that it is possible to calculate force values, which acts to the patient body in the function of the expander angle to the transverse plane. It is shown in all figures that the transverse plane is automatically specified by the model. It is enough to measure one anthropometric patient’s parameter, which is a COG distance from the ground. This input caused the model recalculated the data and the angles were shown in the figures on the axis and were available in the form of the numeric data. (Figure 3, Figure 4 and Figure 5).

## 6. Conclusions

Future studies should take into account the development of a database to help the physiotherapist to first determine the location of expanders depending on the type of disease. Besides, the system should be equipped with dynamometers, which will provide real-time data on patient load. In the future, it will be necessary to use biofeedback, which will enable the patient to actively participate in the rehabilitation process.

There are two major limitations in this study that could be addressed in future research. The postural adjustment can sometimes cause voltage changes that are invisible to the naked eye but can be observed palpationally or thanks to electromyography [18]. When SPIDER cord configuration and electromyography measurements would be correlated, the full condition of patient muscles in situ during the rehabilitation would be known. The second limitation concerns the static parameters. By using inertial measurement units, we could receive the information about patient body acceleration during therapy and we could analyze kinetic parameters [21,22]. Finally, the physical therapy by using SPIDER and proposed model may be beneficial for patients and physiotherapist.

## Figures and Tables

**Figure 1 jpm-10-00033-f001:**
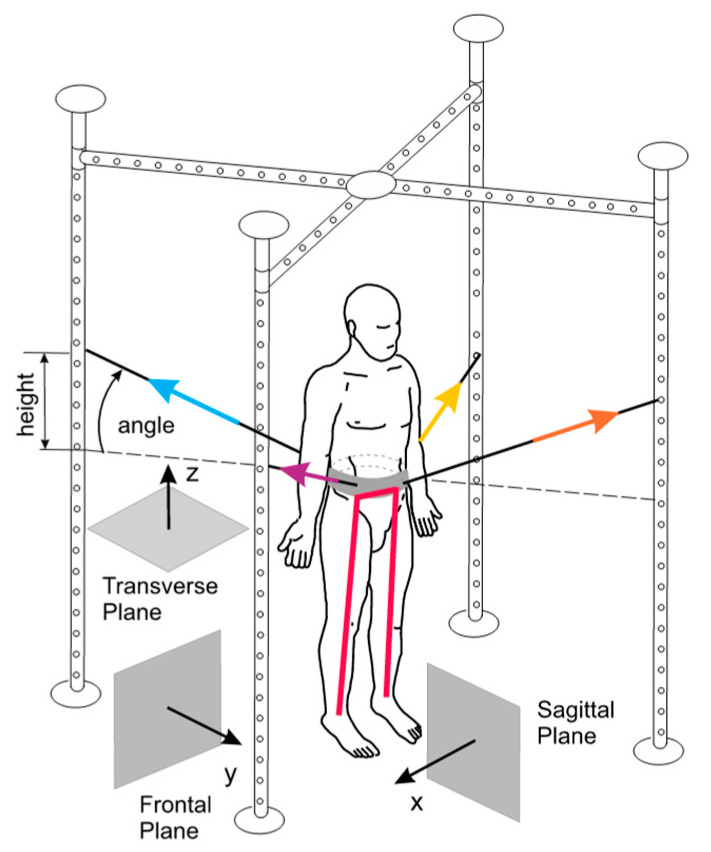
SPIDER (strengthening program of intensive developmental exercises and activities for reaching maximal potential) equipment (SPIDER net) with the reference planes, the expander forces in the standard anatomical position.

**Figure 2 jpm-10-00033-f002:**
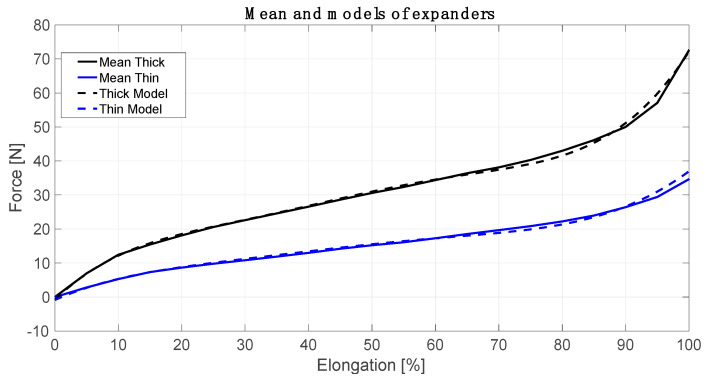
Mean and models of expanders characteristics.

**Figure 3 jpm-10-00033-f003:**
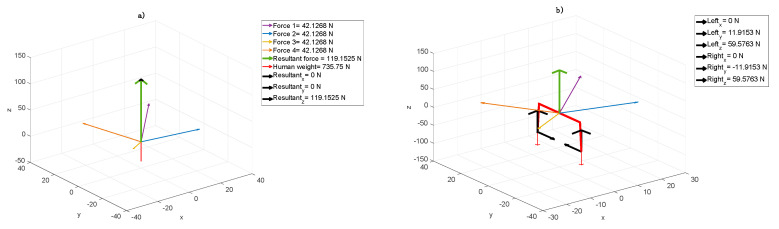
Elastic cord force values in the SPIDER cage, in case of a human with a mass of 75 kg and the following setup: (**a**) elongation cords from 1 to 4—250 mm, and cord and centre of gravity (COG) level angles from 1 to 4—45°; (**b**) elastic cord forces and foot reaction values in the *xyz* directions.

**Figure 4 jpm-10-00033-f004:**
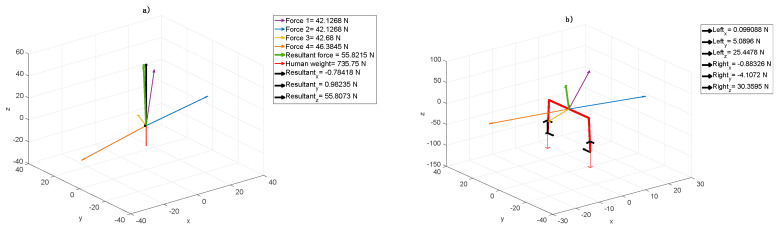
Elastic cord force values in the SPIDER cage in case of a human with a mass of 75 kg and the following setup: (**a**) cord elongations 1 and 2—250 mm, 3—240 mm, and 4—310 mm; COG level angle cords 1 and 2—45°, 3—6°, and 4—48°; (**b**) elastic cord forces and foot reaction values in the *xyz* directions.

**Figure 5 jpm-10-00033-f005:**
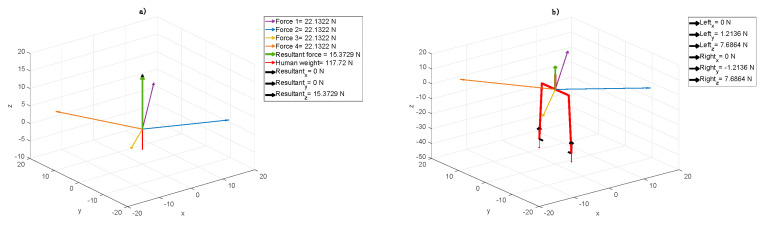
Elastic cord force values in SPIDER cage in case of human mass 12 kg and following set up: (**a**) cord elongations 1,2,3,4–100 mm; cord and COG level angle 1,2,3,4–10 deg, (**b**) elastic cord forces and foot reaction values in XYZ directions.

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
