# Peer review of "SPIDER as A Rehabilitation Tool for Patients with Neurological Disabilities: The Preliminary Research"

_jpm, 2020, doi:10.3390/jpm10020033_

Round 1

Reviewer 1 Report

In present paper, the study concerns - SPIDER as a rehabilitation tool for patients with neurological disabilities.

The approach of the study appears very original. The contents of the manuscript are quite interesting by his methodology and through the tools used.

The manuscript reads smoothly and is easy to understand. The aims, scope, and results of the study are clearly stated. I have very much enjoyed reading this paper. I find it interesting and clearly written, and satisfying also all the other publication criteria of this journal. The study provides a very valuable addition to this line of research, and adds relevantly to the subject with additional original findings. I recommend the publication of this paper after the English corrections by native. Please also extend the literature review and present how the results obtained compare with current literature.

Reviewer 2 Report

My initial reaction was of concern as the authors have clearly not followed the template - the formatting of the title is incorrect, three should be no blank lines between paragraphs, the spacing above the abstract is incorrect, the presentation of the abstract is just incorrect, some of the figure captions are incorrect, there are clearly issues with the reference list.

Upon reading the paper, the first line of the abstract is of concern - "To describe the use of SPIDER device and Matlab software" - this is not the objective of a research paper - that is the objective of a user manual.

Research questions should NOT be in the abstract - the abstract should state what was done in the paper. Likewise - "what is the expander used in SPIDER cage characteristics?" but the expander is not known to this point and SPIDER has not yet been defined. The structure is rather strange, please follow normal paper structure. Likewise "The Matlab software" but what Matlab software and what has this to do with research? Surely the purpose of the paper is to present the model, why is it novel, and validate the model. It does not matter what platform the model was developed on.

"was invented 39 in Russia in 1994 [3]" - what does it matter where it was invented?

"In the following article, the computer model is proposed, " - surely this should be "In the following article, a computer model is proposed, ". There are many computer models in the world. Where did the model come from? Why is it in this form? Why is it different that other models?

The literature survey is missing. There is some discussion in the introduction but not covering the whole research area – only seems to mention in passing.

There seems to be no mention of an ethics approval.

Figures 3, 4, and 5 are too small and poor quality to be read.

The conclusions section starts with "To summerise" - but this is section is a conclusion not a summary! They are different.

Remove text from the paper: "please turn to the CRediT taxonomy for the term explanation. Authorship must be limited to those who have contributed substantially to the work reported."

Overall, it seems there are issues are the preparation, and presentation of the paper. I would propose to rewrite the paper from scratch using the correct template and follow normal paper structures. I would propose to add a full literature search and a literature search section to actually discuss the literature otherwise state of the art cannot be demonstrated and hence novelty cannot be shown (this paper only has 12 references and some of them are not research articles). I would propose to the authors should explicitly state what are the research outcomes from the work.

Round 2

Reviewer 2 Report

 The authors have done a good job and have addressed all of my concerns.